

# Performance in mortality prediction of SAPS 3 And MPM-III scores among adult patients admitted to the ICU of a private tertiary referral hospital in Tanzania: a retrospective cohort study

Nadeem Kassam[1], Eric Aghan[2], Samina Somji[1], Omar Aziz[1], James Orwa[3] and Salim R. Surani[4]

[1] Internal Medicine, Aga Khan University, Dar-es-Salaam, Tanzania
[2] Family Medicine, Aga Khan University, Dar-es-Salaam, Tanzania
[3] Population Health, Aga Khan University, Nairobi, Kenya
[4] Medicine & Pharmacy, Texas A&M University, Texas, United States of America

Corresponding author
Nadeem Kassam,
nadeem.kassam@aku.edu

## ABSTRACT

**Background**. Illness predictive scoring systems are significant and meaningful adjuncts of patient management in the Intensive Care Unit (ICU). They assist in predicting patient outcomes, improve clinical decision making and provide insight into the effectiveness of care and management of patients while optimizing the use of hospital resources. We evaluated mortality predictive performance of Simplified Acute Physiology Score (SAPS 3) and Mortality Probability Models ($MPM_0$-III) and compared their performance in predicting outcome as well as identifying disease pattern and factors associated with increased mortality.

**Methods**. This was a retrospective cohort study of adult patients admitted to the ICU of the Aga Khan Hospital, Dar-es-Salaam, Tanzania between August 2018 and April 2020. Demographics, clinical characteristics, outcomes, source of admission, primary admission category, length of stay and the support provided with the worst physiological data within the first hour of ICU admission were extracted. SAPS 3 and $MPM_0$-III scores were calculated using an online web-based calculator. The performance of each model was assessed by discrimination and calibration. Discrimination between survivors and non–survivors was assessed by the area under the receiver operator characteristic curve (ROC) and calibration was estimated using the Hosmer-Lemeshow goodness-of-fit test.

**Results**. A total of 331 patients were enrolled in the study with a median age of 58 years (IQR 43-71), most of whom were male ($n = 208$, 62.8%), of African origin ($n = 178$, 53.8%) and admitted from the emergency department ($n = 306$, 92.4%). In-hospital mortality of critically ill patients was 16.1%. Discrimination was very good for all models, the area under the receiver-operating characteristic (ROC) curve for SAPS 3 and $MPM_0$-III was 0.89 (95% CI [0.844–0.935]) and 0.90 (95% CI [0.864–0.944]) respectively. Calibration as calculated by Hosmer-Lemeshow goodness-of-fit test showed good calibration for SAPS 3 and $MPM_0$-III with Chi-square values of 4.61 and 5.08 respectively and $P$–Value > 0.05.

**Conclusion**. Both SAPS 3 and $MPM_0$-III performed well in predicting mortality and outcome in our cohort of patients admitted to the intensive care unit of a private tertiary

hospital. The in-hospital mortality of critically ill patients was lower compared to studies done in other intensive care units in tertiary referral hospitals within Tanzania.

## BACKGROUND

The burden of critical care and ICU mortality is greatest in countries with low global national income (*Vincent et al., 2014*). The reported ICU mortality widely varies from one setting to the other with higher rates reported in low and middle-income countries (LMICs) (*Vincent et al., 2014*; *Ilori & Kalu, 2012*; *Smith, Ayele & McDonald, 2013*). As of 1st July 2020, the World Bank upgraded Tanzania's economic status from a low to lower-middle income country due to its strong economic performance over the past decade. However, the availability of intensive care units in Tanzania is very limited; none of the seven district hospitals surveyed in 2009 had an ICU. The four national referral hospitals had a total of only 38 ICU beds serving a population of 57 million (*Baker et al., 2013*). This is in contrast to high-income countries (HICs) which generally have between five to 30 ICU beds per 100,000 people (*Dondorp, Iyer & Schultz, 2016*). The availability and improvement of quality of care of critical illness in LMICs is necessary to reduce this burden and even more significant in the coming years as the population ages and prevalence of comorbidities increases (*Adhikari et al., 2010*).

Despite the use of high-cost and sophisticated devices, ICU mortality rates remain high. The burden of diseases compounded by a severe lack of resources, specialists and data makes prediction of ICU outcomes in terms of morbidity and mortality a crucial component of care across the continent. In HICs mortality prediction models are not only used to predict outcome but also as tools for quality enhancement and analytical decision making. These mortality predictive models were developed more than 25 years ago using patient characteristics. They help quantify the severity of illness, estimate the gravity of the disease, help predict outcome and assist in resource allocation (*Keegan, Gajic & Afessa, 2011*; *Zimmerman et al., 1995*).

The three major predictive scoring systems used to predict mortality in general ICU patients are the Acute Physiologic and Chronic Health Evaluation (APACHE) scoring system, the Simplified Acute Physiologic Score (SAPS) and the Mortality Prediction Model ($MPM_0$) (*Juneja et al., 2012*). APACHE-IV, SAPS 3, and $MPM_0$-III are the latest versions of the aforementioned scoring systems (*Zimmerman et al., 2006*; *Metnitz et al., 2005*; *Higgins et al., 2007*). When selecting a predictive scoring system for use in a given ICU, it is essential to use a model that is well-proven, established and validated contextually. APACHE-IV has long been considered more precise for predicting mortality but is perceived as burdensome and more costly especially in resource-limited settings (*Kuzniewicz et al., 2008*). $MPM_0$-III is considered superior in resource-limited settings since it has the lowest extraction burden among the three models and is available without cost on various medical information

sites. $MPM_0$-III has been a well-studied tool in East Africa. Nevertheless, its performance in predicting outcomes among critically ill patients admitted to ICU's in Kenya (*Lukoko et al., 2020*) and Rwanda (*Riviello et al., 2016*) are contradictory. External validation of SAPS 3 among patients admitted to the ICU's in Austria, Brazil and Italy reported that SAPS 3 had a good ability to predict outcomes but performed poorly across all probabilities of death when compared with APACHE-IV and $MPM_0$-III (*Poole et al., 2009*; *Nassar Jr et al., 2012*; *Metnitz et al., 2009*). SAPS 3 may have greater potential for international use since the score was derived from data in more than one country (*Metnitz et al., 2005*). No study to date has assessed the performance of SAPS 3 in LMICs, especially in Sub-Saharan Africa.

These aforementioned predictive scoring systems have been compared in different studies and have produced variable results. The existence of a large number of scoring systems with contrasting performance suggests the best fit model is ICU specific. Thus, each particular ICU needs to determine which scoring system performs best in its setup; hence there was a need to carry out a comparative study in our cohort of patients to identify the best performing model. There have been no studies done in Tanzania that have compared the performance of these scoring models. The study had two main objectives: (1) To compare the performance of $MPM_0$-III and SAPS 3 in order to identify which model best fits in the ICU of the Aga Khan Hospital, Dar es Salaam. (2) To identify disease patterns and risk factors associated with higher mortality rates among critically ill patients.

## METHODS

This was a single centre retrospective cohort study, conducted in the ICU of the Aga Khan Hospital, Dar-es- Salaam, Tanzania. The Aga Khan Hospital is the largest private hospital and the only Joint Commission International (JCI) accredited hospital in Tanzania. The ICU of the Aga Khan Hospital is a 15 bed unit which provides level III services to all kind of critically ill patients. The ICU is capable of providing mechanical ventilation, inotropic support and renal replacement therapy. The unit is divided in 3 sections –7 adult general ICU beds (including 2 isolation rooms), 4 for cardiac patients and 4 for paediatrics. The ICU is an open-model one run by a multidisciplinary team comprising of physician of the primary specialty of care, physiotherapist and dietician led by a full-time critical care specialist. The nurse-to-patient ratio ranges between 1:1 and 1:2. All adult patients aged 18 and above admitted to the ICU were eligible for the study. Patients admitted for observation, having incomplete data and those whose duration of stay in the unit was less than an hour as well as those diagnosed with COVID–19 were excluded from the study. Admissions to the ICU are limited to those meeting a strict admitting criteria set by the hospital. A total of 747 adults patients were admitted to the ICU from August 2018 to April 2020. A sample size of 331 patients having a specific outcome (death or discharge home) was determined to be sufficient to give the study a 80% power and 95% confidence for detection of 10% difference in performance between SAPS 3 and $MPM_0$-III. The ICU admission register was used to identify patients admitted and patient files were retrieved from the medical records. The medical file numbers were entered into a computer and

computer generated random sampling was performed until the desired sample size was achieved. Patient demographics and clinical data were extracted using patient records and were entered into a spreadsheet on Microsoft Office Excel 2010 (Redmond, WA, USA). Data was extracted by experienced junior doctors with working experience in the ICU and was independently verified by the primary author for accuracy and completeness. The reasons for admission were grouped into 11 categories: surgery, gastroenterology, neurology, endocrinology, respiratory, cardiovascular, nephrology, sepsis, oncology, hematology, obstetrics and gynecology. When multiple diagnoses were present, the leading one with the worst prognosis was selected as the main reason for admission.

Descriptive analysis of demographic characteristics were done and presented as percentages while the categorical and continuous outcome variables were analysed and presented as means and medians with interquartile ranges respectively. Categorical and continuous variables between survivors and non-survivors were compared using Pearson's chi-square test and Mann–Whitney $U$ test respectively. SAPS-3 and $MPM_0$-III were calculated using an online scoring calculator, available on http://www.uptodate.com/. Accurate discrimination and calibration are key distinguishing features that should be met by all predictive scoring models. Discrimination of the model was assessed by the area under the receiver operating characteristic (ROC) curve. An area of 0.7–0.8 is reflected as fair, 0.8–0.9 good and >0.9 excellent. Non-parametric Wilcoxon statistics was used to compare the area under the ROC curves (*Steyerberg et al., 2010*). A Hosmer-Lemeshow goodness-of-fit test which follows chi-square distribution was used to evaluate the model fit as well as calibration of the models with a $p$-value of >0.05 signifying no evidence of poor fit (*Steyerberg et al., 2010*). However, all other statistical tests with a $p$-value of <0.05 were considered statistically significant. Any variable with $P$-Value <0.05 and those considered clinically significant in explaining mortality in the ICU were considered in multivariable model. Determinants of mortality among critically ill patients were identified using binary logistic regression; odds ratio with corresponding 95% Confidence intervals (CI) and $P$-value were reported. All statistical analysis was done using STATA version 15. The study protocol was approved by the Ethical Research Committee (ERC) of the Aga Khan University (AKU/2020/051/fb) and individual consent of each study participant was exempted since it did not affect the rights and welfare of the patients.

## RESULTS

A total of 331 patients were included in the study. Out of the 331 patients ($n = 278$, 83.9%) survived and ($n = 53$, 16.1%) died. Table 1 below shows general and clinical characteristic of the cohort and provides a comparison of survivors to non-survivors. The median age of the cohort was 58 years (IQR 43-71) with more than half of admitted patients being male ($n = 208$, 62.8%) and of African origin ($n = 178$, 53.8%). Most of the patients were admitted to the ICU from the emergency department ($n = 306$, 92.5%), who were at home prior ($n = 318$, 96.1%) with majority of them suffering from neurological disease ($n = 63$, 19%), sepsis ($n = 60$, 18.1%), respiratory ($n = 36$, 10.9%) and cardiovascular ($n = 36$, 10.9%) related conditions. Median ICU and hospital LOS were 4

**Table 1 General and clinical characteristics of patients ($N = 331$).**

| Characteristics | All ($N = 331$) | Survivors ($n = 278$) | Non-survivors ($n = 53$) | p-value |
|---|---|---|---|---|
| | N (%) | n (%) | n (%) | |
| **Sex** | | | | |
| Male | 208(62.8) | 174(62.6) | 34(64.2) | 0.829 |
| Female | 123(37.2) | 104(37.4) | 19(35.8) | |
| **Age group in years** | | | | |
| <45 | 87(26.3) | 79(28.4) | 8(15.1) | |
| 45–64 | 114(34.4) | 99(35.6) | 15(28.3) | |
| 65–74 | 61(18.4) | 51(18.4) | 10(18.9) | 0.015 |
| 75–84 | 51(15.4) | 36(12.9) | 15(28.3) | |
| >84 | 18(5.4) | 13(4.7) | 5(9.4) | |
| **Ethnicity** | | | | |
| African | 178(53.8) | 152(54.7) | 26(49.1) | |
| Asian | 136(41.1) | 111(39.9) | 25(47.2) | 0.589 |
| Other | 17(5.1) | 15(5.4) | 2(3.8) | |
| **Admitted from** | | | | |
| Emergency | 306(92.5) | 257(92.5) | 49(92.5) | |
| Wards | 15(4.5) | 12(4.3) | 3(5.7) | 0.800 |
| Clinic | 10(3.0) | 9(3.2) | 1(1.9) | |
| **Location before ICU admission** | | | | |
| Home | 318(96.1) | 267(96.0) | 51(96.2) | 0.950 |
| Hospital | 13(3.9) | 11(4.0) | 2(3.8) | |
| **Admitting Category** | | | | 0.014 |
| Surgery | 38(11.5) | 36(12.9) | 2(3.8) | |
| Gastroenterology | 31(9.4) | 25(9.0) | 6(11.3) | |
| Neurology | 63(19.0) | 53(19.1) | 10(18.9) | |
| Endocrinology | 18(5.4) | 17(6.1) | 1(1.9) | |
| Respiratory | 36(10.9) | 35(12.6) | 1(1.9) | |
| Cardiovascular | 36(10.9) | 30(10.8) | 6(11.3) | |
| Nephrology | 16(4.8) | 12(4.3) | 4(7.6) | |
| Sepsis | 60(18.1) | 47(16.9) | 13(24.5) | |
| Obstetrics and Gynecology | 10(2.0) | 9(3.2) | 1(1.9) | |
| Hematology | 7(2.1) | 5(71.4) | 2(3.8) | |
| Oncology | 16(4.8) | 9(3.2) | 7(13.2) | |
| **Code Status on Admission** | | | | |
| DNR | 40(12.1) | 13(4.7) | 27(50.9) | <0.001 |
| Full Code | 291(87.9) | 265(95.3) | 26(49.1) | |
| **Age in years** | **58(43–71)[a]** | **55.5(41–70)[a]** | **70(55–78)[a]** | **0.0003[b]** |
| **SAPS 3 scores** | **42(32–51)[a]** | **39(31–48)[a]** | **60(51-68)[a]** | **<0.0001[b]** |
| **MPM$_0$III scores** | **3(2–4)[a]** | **3(2–4)[a]** | **5(4–6)[a]** | **<0.0001[b]** |

| Characteristics | All (N = 331) | Survivors (n = 278) | Non-survivors (n = 53) | p-value |
|---|---|---|---|---|
| | N (%) | n (%) | n (%) | |
| LOS ICU (days) | 4(2–6)[a] | 4(2–6)[a] | 6(2–11)[a] | 0.0029[b] |
| LOS Hospital (days) | 6(4–10)[a] | 6(4–10)[a] | 8(3–13)[a] | 0.3248[b] |

Notes.

[a] median (IQR)

[b] p-value for Mann–Whitney U test.

LOS, Length of Stay; DNR, Do Not resuscitate.

Data in median (IQR), and n (%).

(*Ilori & Kalu, 2012*; *Smith, Ayele & McDonald, 2013*; *Baker et al., 2013*; *Dondorp, Iyer & Schultz, 2016*; *Adhikari et al., 2010*) and 6 (*Baker et al., 2013*; *Dondorp, Iyer & Schultz, 2016*; *Adhikari et al., 2010*; *Keegan, Gajic & Afessa, 2011*; *Zimmerman et al., 1995*; *Juneja et al., 2012*; *Zimmerman et al., 2006*) days respectively.

When survivors and non-survivors were compared, there was a statistically significant difference ($P$-value <0.05) in age, length of ICU stay, admitting category and code status. Higher mortality rates were noted in the elderly aged between 75–84 years ($n = 15$, 28.3%), those suffering from sepsis ($n = 13$, 24.5%) and neurological diseases ($n = 10$, 18.9%). Higher percentage of mortality was also noted in Do-Not-Resuscitate (DNR) patients ($n = 27$, 50.9%) compared to those without limitations of care. There was no statistically significant difference between survivors and non-survivors by sex, ethnicity and prior location before ICU admission. The overall SAPS 3 score for all the patients was 42 (IQR: 32–51) of which non-survivors had a higher score 60 (IQR: 51–68) than the survivors 39 (IQR: 31–48) with $p$-value (<0.0001). Similarly, the median $MPM_0$-III score in non-survivors, 5 (IQR: 4–6) was higher than survivors, 3 (IQR: 2–4) with $p$-value (<0.0001).

Table 2 below, illustrates the type of support patients received in the first 24 h of ICU admission. Of the 331 patients admitted to the ICU, ($n = 123$, 37.2%) patients received support in the first hour of ICU admission that included: mechanical ventilation, inotropes and hemodialysis. Highest mortality was noted amongst those who required inotropic support and mechanical ventilation ($n = 21$, 39.6%) in the first hour of ICU admission.

Table 3 below highlights comorbid conditions amongst the critical ill patients admitted to the ICU. More than one comorbid condition per critically ill patient was recorded when present. The most common comorbid condition amongst our cohort was hypertension ($n = 174$, 52.6%) and diabetes mellitus ($n = 107$, 32.3%).When comorbid conditions were compared between survivors and non- survivors, a higher percentage of mortality with statistically significant difference ($P$ value <0.05) was noted among those suffering with chronic kidney disease ($n = 14$, 26.4%) and liver cirrhosis ($n = 7$, 13.2%).

Calibration of each scoring system exhibited good performance. The goodness of fit Hosmer-Lemeshow test and $p$-value of each scoring system is shown in Table 4 below. The area under ROC of SAPS 3 and $MPM_0$-III in prediction of mortality are shown below in Fig. 1 below. The area under the ROC was calculated to evaluate the predictive value of the scoring systems. The area under the ROC curve for the SAPS 3 showed a statistically

**Table 2  Type of support received in the first hour of ICU admission.**

| Type of support | All (N = 331) | Survivors (n = 278) | Non-survivors (n = 53) | p-value |
|---|---|---|---|---|
| | N (%) | n (%) | n (%) | |
| None | 208(62.8) | 196(70.5) | 12(22.6) | <0.001 |
| Hemodialysis | 13(3.9) | 11(4.0) | 2(3.8) | 0.9498 |
| Inotropes | 33(10.0) | 29(10.4) | 4(7.6) | 0.5206 |
| Mechanical ventilation | 26(7.9) | 17(6.1) | 9(17.0) | 0.0070 |
| Inotropes, Hemodialysis | 5(1.5) | 5(1.8) | 0(0.0) | 0.3252 |
| Inotropes, mechanical ventilation | 36(10.9) | 15(5.4) | 21(39.6) | <0.001 |
| Mechanical ventilation, Hemodialysis | 3(0.9) | 3(1.1) | 0(0.0) | 0.4474 |
| Inotropes, mechanical ventilation, hemodialysis | 7(2.1) | 2(0.7) | 5(9.4) | 0.0001 |

**Notes.**
Data presented in n (%).

**Table 3  Comorbid conditions among critically ill patients admitted to the ICU.**

| Comorbidity | All (N = 331) | Survivors (n = 278) | Non-survivors (n = 53) | p-value |
|---|---|---|---|---|
| | N (%) | n (%) | n (%) | |
| Hypertension | 174(52.6) | 146(52.5) | 28(52.8) | 0.9553 |
| Diabetes Mellitus | 107(32.3) | 87(31.3) | 20(37.7) | 0.3613 |
| Heart Failure | 53(16.0) | 41(14.7) | 12(22.6) | 0.1510 |
| Chronic Kidney Disease | 45(13.6) | 31(11.2) | 14(26.4) | 0.0030 |
| HIV | 21(6.3) | 16(5.8) | 5(9.4) | 0.314 |
| COPD | 17(5.1) | 15(5.4) | 2(3.8) | 0.6239 |
| CAD | 17(5.1) | 15(5.4) | 2(3.8) | 0.6239 |
| Liver Cirrhosis | 16(4.8) | 9(3.2) | 7(13.2) | 0.0022 |
| DM & HTN | 87 (26.3) | 72 (25.9) | 15 (28.3) | 0.7157 |
| HTN & CKD | 39 (11.8) | 27 (9.7) | 12 (22.6) | 0.0075 |
| DM & HTN & CKD | 30 (9.1) | 21 (7.6) | 9 (17.0) | 0.0285 |

**Notes.**
Data presented in n (%).

significant predictive marker of mortality (AUC: 0.8892; Severity of illness scoring syst CI [0.844–0.935]). The cut-off value for SAPS 3 was 54 with the sensitivity of 72% and specificity of 90%. The $MPM_0$-III scoring system also showed a statistically significant predictive marker for the outcome of interest (AUC: 0.904; 95% CI [0.864–0.944]). The cut-off value for $MPM_0$-III was 4, with sensitivity of 74% and specificity of 87%. There was no statistically significant difference between the ROC curves of the two models (P-value = 0.2418) (Table 5).

**Table 4** Goodness of fit Hosmer-Lemeshow test and *p*-value of each scoring model.

| Scoring Model | Chi –Square | P-Value |
|---|---|---|
| MPM 0- III Score | 5.08 | 0.2791 |
| SAPS 3 | 4.61 | 0.7980 |

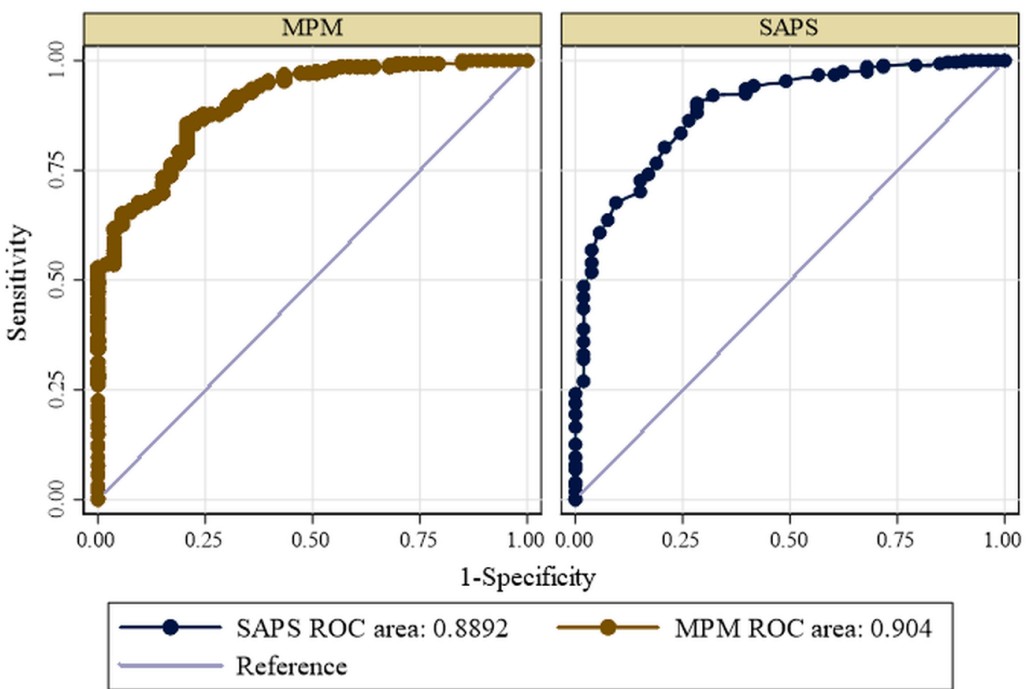

**Figure 1** Receiver operating curve for predicting mortality according to SAPS 3 and $MPM_0$ III models.

**Table 5** Area under curve and 95% confidence Intervals for the models.

| Variable | Cut-off | AUC | LL | UL | P-value |
|---|---|---|---|---|---|
| MPM 0- III Score | 4 | 0.904 | 0.864 | 0.944 | 0.2418 |
| SAPS 3 | 54 | 0.8892 | 0.8440 | 0.935 | |

**Notes.**
LL, Lower limits; UL, Upper limits; AUC, Area under the ROC curve.

The overall estimated median (IQR) predicted mortality among the 331 ICU patients was 6% (2%–20%) on the SAPS 3 model and 11.5% (3.8%–27.9%) based on the $MPM_0$-III model. The stratified analysis by survivors and non-survivors is shown in Fig. 2 below. The median predicted mortality risk for survivors is lower than those of non-survivors. In the SAPS 3 model, the estimated median for survivors was 5% (IQR: 1%–11%) while for the non-survivors this was 50% (IQR: 34%–69%) Based on the $MPM_0$-III model the median

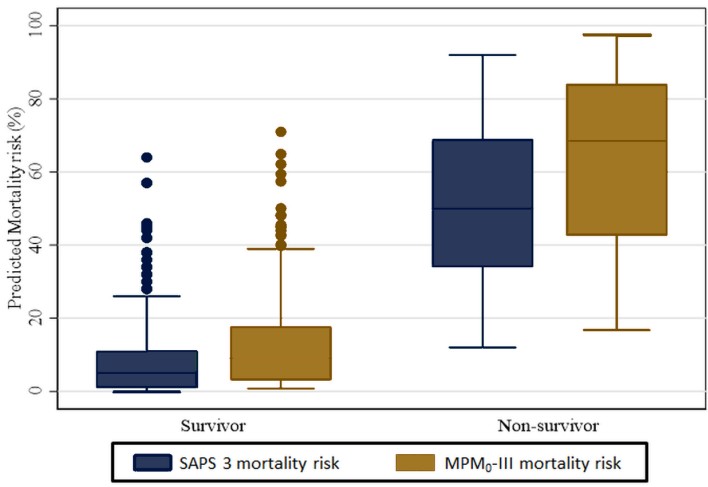

**Figure 2** Median predicted mortality rates for SAPS 3 and MPM$_0$-III.

**Table 6** Factors associated with increased odds of mortality among critically ill patients.

| Characteristics | Unadjusted odds ratio | | | Adjusted odds ratio | | |
|---|---|---|---|---|---|---|
| | OR | 95% CI | *P*-value | OR | 95% CI | *P*-value |
| **Age in years** | 1.032 | 1.014-1.051 | 0.001 | 1.020 | 0.997–1.043 | 0.086 |
| **LOS in ICU** | 1.068 | 1.021-1.117 | <0.001 | 1.462 | 1.179–1.814 | 0.001 |
| **LOS in hospital** | 1.017 | 0.994-1.041 | 0.141 | 0.717 | 0.580–0.886 | 0.002 |
| **Sex** | | | | | | |
| Male | ref | | | | | |
| Female | 0.935 | 0.507-1.723 | 0.829 | 0.870 | 0.399–1.893 | 0.725 |
| **Admitted From** | | | | | | |
| Emergency | ref | | | | | |
| Wards | 1.311 | 0.357-4.819 | 0.683 | 5.341 | 1.278–22.322 | 0.022 |
| Clinic | 0.583 | 0.072-4.704 | 0.612 | 1.033 | 0.114–9.347 | 0.977 |
| **Code status** | | | | | | |
| DNR | ref | | | | | |
| Full code | 0.047 | 0.022-0.102 | <0.001 | 0.052 | 0.021–0.129 | <0.001 |

predicted mortality was 9.1% (3.1%−1.7%) and 68.5% (IQR: 42.7%–84.0%) for survivors and non-survivors respectively.

Multiple clinical factors were associated with increased adjusted odds of mortality. These included length of ICU stay (adjusted odds ratio [aOR], 1.462; *P*-Value = 0.001) and those transferred from the ward (aOR, 5.341; *P*-value<0.022). However, it was protective to stay longer in the hospital as the odds of mortality decreased as the length of hospitalization increased (aOR, 0.717; *P*-Value = 0.002) (Table 6).

## DISCUSSION

To our knowledge, this is the first study to report on performance of predictive scoring models in Tanzania and more so in a private setting. Accurate discrimination and calibration are two key characteristics that should be met by all predictive scoring systems. Both SAPS 3 and $MPM_0$-III performed well in our cohort. According to our results, a SAPS 3 score of higher than 54 can predict mortality with sensitivity of 72% and specificity of 90%. A $MPM_0$-III score of greater than 4 can predict mortality with sensitivity of 74% and specificity of 87%.

Discrimination describes the accuracy of a given prediction in our cohort, the discriminatory capability of both SAPS 3 (20 variables) and $MPM_0$-III (16 variables) was good. There was no statistically significance difference when both these models were compared, suggesting that the model with more variables was not associated with better discriminatory performance. $MPM_0$-III has been externally validated in various ICUs in North America (*Higgins et al., 2007*; *Kuzniewicz et al., 2008*; *Higgins et al., 2009*) and has shown to have good discrimination which was similar to our study finding. However, a study done at Aga Khan University Hospital, Nairobi, Kenya (*Lukoko et al., 2020*) and two public ICUs in Rwanda (*Riviello et al., 2016*) showed $MPM_0$-III to have fair discrimination amongst their cohort. This observed difference in discrimination may be due to the effect of differences in proportion of case mixes between the study settings. Similarly SAPS 3 has been externally validated in various ICUs in Italy (*Poole et al., 2009*), Brazil (*Nassar Jr et al., 2012*), Austria (*Metnitz et al., 2009*) and found to have good discriminatory capability amongst their cohort. Despite SAPS 3 having greater prospective for international generalizability there has been no published studies evaluating its performance in Sub-Saharan African ICUs. This is the first study that reports its potential for application in LMICs.

Calibration describes how the instrument performs over a wide range of predicted mortalities. Calibration is sensitive to alterations in case-mix and patient care and interventions. Despite its tendency to deteriorate over time and leading to overestimation of mortality (*Nassar Jr et al., 2012*), both SAPS 3 and $MPM_0$-III were well calibrated amongst the critically ill patients admitted at our study setting. Our study findings were contrary to SAPS 3 validation studies mentioned earlier which reported poor calibration and overestimation of mortality (*Poole et al., 2009*; *Nassar Jr et al., 2012*; *Metnitz et al., 2009*). However, external validation studies have reported $MPM_0$-III to have good calibration (*Higgins et al., 2007*; *Kuzniewicz et al., 2008*; *Higgins et al., 2009*). Earlier studies mentioned that were conducted in Sub-Saharan Africa have produced contrasting results. The $MPM_0$-III was well calibrated amongst the critical ill patients admitted to the ICU of the Aga Khan University Hospital, Nairobi (*Lukoko et al., 2020*) but showed poor calibration amongst all adult patients admitted to Rwanda's two public ICUs (*Riviello et al., 2016*). These findings highlight the similar treatment protocols and interventions between two sister hospitals located in different geographical regions.

In this retrospective study we also aimed to identify patient demographics, disease patterns, clinical outcomes as well as factors associated with higher risk of mortality in

patients admitted to the ICU of the Aga Khan Hospital, Dar –es Salaam. Based on this retrospective observational cohort, the in-hospital mortality of critically ill patients was 16.1%, which is far less than the reported mortality among all other tertiary referral hospitals in Tanzania, 41.4% (*Sawe et al., 2014*) but slightly exceeds rates reported in western Europe and North America (*Vincent et al., 2014*). This disparity is not surprising since the intensive care unit at our setting has access to more resources than similar units in the country and comparable in various ways to facilities in HICs. The ICU cohort studied in the four tertiary referral hospitals in Tanzania was younger (median age 34 years, IQR 21-53) compared to our study population (median age 58 years, IQR 43–71). This variation could be due to the exclusion of patients aged less than 18 years in our study. However both the cohorts had male predominance of 57.5% and 62.8% respectively (*Sawe et al., 2014*). The bulk of admissions in our cohort were those suffering from neurological disease, sepsis, respiratory and cardiovascular related conditions. Mortality was highest among those admitted due to sepsis. Our results are in parallel with a large intercontinental database that emphasized the association of sepsis with high mortality rates in all countries (*Vincent et al., 2014*). The median length of ICU stay is similar to reports from tertiary hospitals in Sub-Saharan Africa (*Sawe et al., 2014*; *Kwizera, Dunser & Nakibuuka, 2012*).

Prolonged length of stay (LoS) in the ICU and patients transferred from the general ward to the ICU were factors associated with higher adjusted odds of mortality among critically ill patients. Prolonged LoS in the ICU may be attributed to development of multi- systemic complications necessitating continued organ support. There are no laws and guidelines in Tanzania with regards to withdrawal of support, hence we hypothesize that significant fraction of patients with a prolonged course of illness and with expected poor outcomes are admitted for extended intervals before succumbing to death. Our study findings are comparable to several studies done in well-equipped ICUs that concluded patients with multiple diseases and having organ dysfunction were key factors that prolong the LoS in ICU (*Toptas et al., 2018*; *Moitra et al., 2016*). Contrasting results have also been published that LoS in ICU is not an independent risk factor for in-hospital mortality (*Williams et al., 2010*). Those patients transferred from the general ward to the ICU also had higher adjusted odds of mortality; this is not surprising since it is a mere reflection of deteriorating physiological and clinical condition. Few studies have demonstrated early transfer to the ICU for treatment to have a substantial impact on in-hospital mortality and LoS (*Churpek et al., 2016*; *Young et al., 2003*).

We identified several limitations in our study. Firstly, this was a single center study and as such the findings may not be valid across all patient populations in Tanzania. Secondly, since our study was a retrospective design it restricted us the ability to follow up outcomes after ICU discharge and doesn't provide the same level of evidence as a prospective study design.

## CONCLUSION

In summary, this is the first and largest study to report on performance of predictive scoring models in Tanzania. Our study concluded both SAPS 3 and $MPM_0$-III performed well in

predicting mortality well among critically ill patients admitted to the ICU of the Aga Khan Hospital, Dar es Salaam, Tanzania. We found our mortality among critically ill patients to be much lower compared to other tertiary referral hospitals in Tanzania. Amongst our cohort, patients with sepsis had the highest mortality rate. Thus clinical research targeting infection prevention efforts and early implementation of targeted interventions would be important to improving outcomes. Prolonged ICU stay and transfer from general wards to ICU were key factors of mortality. Of note, the performance of predictive scoring models tend to deteriorate over time; termed as worsening of discrimination and calibration and resulting in overestimation of mortality (*Nassar Jr et al., 2012*). Thus periodic updating is crucial for sustaining accuracy of these predictive models.

**Abbreviations**

| | |
|---|---|
| **APACHE** | Acute physiology and chronic health evaluation |
| **SAPS** | Simplified acute physiology score |
| **MPM** | Mortality probability models |
| **LoS** | Length of Stay |
| **ICU** | Intensive care Unit |

## ACKNOWLEDGEMENTS

Special thanks to Dr. Kamran Hameed, head of the Internal Medicine department at the Aga Khan University and Hospital, Dar-es-Salaam for his continuous mentorship and guidance during my residency program.

### Funding
The authors received no funding for this work.

### Competing Interests
The authors declare there are no competing interests .

### Author Contributions
- Nadeem Kassam conceived and designed the experiments, performed the experiments, prepared figures and/or tables, authored or reviewed drafts of the paper, and approved the final draft.
- Eric Aghan, Samina Somji, Omar Aziz and Salim R. Surani conceived and designed the experiments, performed the experiments, authored or reviewed drafts of the paper, and approved the final draft.
- James Orwa analyzed the data, prepared figures and/or tables, authored or reviewed drafts of the paper, and approved the final draft.

## Human Ethics

The following information was supplied relating to ethical approvals (i.e., approving body and any reference numbers):

The Study was approved by the Aga Khan University Ethical research committee (AKU- ERC, EA): AKU/2020/051/fb. The National Institute for Medical Research (NIMR) mandates the AKU –ERC to approve health research conducted by Tanzanian staff and students under the Act of Parliament No. 23 of 1979 and its amendments in 1997.

## Ethics

The following information was supplied relating to ethical approvals (i.e., approving body and any reference numbers):

The Study was approved by the Aga Khan University Ethical research committee (AKU- ERC, EA): AKU/2020/051/fb. The National Institute for Medical Research (NIMR) mandates the AKU –ERC to approve health research conducted by Tanzanian staff and students under the Act of Parliament No. 23 of 1979 and its amendments in 1997.

## Data Availability

Raw data is available as a Supplementary File.

## Supplemental Information

Supplemental information for this article can be found online at http://dx.doi.org/10.7717/peerj.12332#supplemental-information.

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
