# Peer review of "Performance in mortality prediction of SAPS 3 And MPM-III scores among adult patients admitted to the ICU of a private tertiary referral hospital in Tanzania: a retrospective cohort study"

_PeerJ, doi:10.7717/peerj.12332_

## Round 0.1 · original submission · Major Revisions

Dear authors,

Please address the issues brought up by the reviewers. Please show that you provide a contribution to the literature by putting the local perspective into perspective.

Additionally, I demand you describe the statistical methods adequately. E.g., the logistic regression fitting and adjustment are not described.

Reviewer 1 ·

Basic reporting

Kassam et al. report on a small single-centre experience with approximately 300 intensive care patients. The aim of the paper is to compare SAPS 3 and MPM 0-III for outcome prediction.

Experimental design

Retrospective cohort study.

Validity of the findings

From my point of view, there is no novelty in this study.

(e.g. https://pubmed.ncbi.nlm.nih.gov/28523584/)

Additional comments

This retrospective analysis does not generate new hypotheses.

Reviewer 2 ·

Basic reporting

The article is mostly written clearly and unambiguously; the phrase "mortality predictive performance" - placed prominently in the title - does seem a bit odd, though. I would suggest phrasing such as "performance in mortality prediction for ease and clarity.

Literature references provided are sufficient to provide adequate background and context.

The manuscript follows the usualarticle structure; figures, tables are well presented; raw data are shared.

The article overall is self-contained with relevant results to hypotheses.

Specific comments:
Line 160ff: Please provide absolute numbers in addition to percentages for clean reporting.

Line 112ff: The usual acronym is "SAPS 3" (rather than "SAPS III") and should be used consistently.

Experimental design

The article includes original primary research with aims and scope I would consider within the stated width of the journal. The research question well defined, relevant and meaningful. It is clearly stated how this research fills an identified knowledge gap.

The investigation has been performed to a technical standard that requires clarification. Ethical standards are adhered to. Methods used are described with insufficient detail as of yet.

Specific comments:
Line 122: "The Intensive care unit (ICU) of the Aga Khan Hospital is 15 bed modern and well-equipped unit which provides level III services to all kind of critically ill patients."- While this is undoubtedly the case, "modern" and "well-equipped" are both ill-defined terms which should not be used in such an absolute manner.

Line 127: "The ICU is run by a multidisciplinary team, comprising of the primary care physician [...]" - Does that indeed mean a primary care physician (i.e., doctors specialising in primary care) or physicians of the primary specialty of care of the respective patient (e.g., surgeons), as in an "open" ICU model? Please clarify.

Line 133ff: "A 133 total of 747 adults patients were admitted to the ICU from August 2018 to April 2020. A sample size of 331 patients was determined to be sufficient to give the study a 80% power and 95% confidence for detection of 10% difference in performance between SAPS-III and MPM0-III." Such a power-calculation is laudable, but larger sample size - especially if available - would certainly be preferable. Indeed, the sample size may be sufficient for score comparison, but is undoubtedly weak for performance assessment of scores in general (without the risk of over- or underfitting).

Line 148ff: I miss any definition of mortality (i.e., the study's endpoint) in this section. Later paragraphs only adress "survivors" and "non-survivors". Line 59 of the hints at "in- intensive care mortality". If that is the case, this either constitutes a flaw in the study's design or needs special attention, as all models cited actually calculare hospital mortality rates rather than ICU mortality rates (with the latter necessarily being lower).

Line 155ff: "A Hosmer-Lemeshow goodness-of-fit test was used to assess the calibration of the models with a p-value > 0.05 considered statistically significant." This is confusing and contrary to my understanding of the Hosmer-Lemeshow-test. Furthermore, the known limitations of this very test must be expressed and emphasised.

Validity of the findings

Line 95: "APACHE-IV has long been considered more precise for predicting mortality than the other scoring systems" - This statement cannot be supported by the given reference 13 (Kuzniewicz et al.), since it does not adress SAPS 3 (but rather SAPS II, which - no matter the name - has been derived independently from SAPS II).

Line 101ff: "External validation in other ICU populations reported that SAPS 3 had good discrimination, but poorer calibration, when compared with APACHE-IV and MPM0-III. However, it may have greater potential for international use since the score was derived from data in more than one country" - While the latter is certainly true, the statement overall cannot hold true, if calibration were off overall. Recalibration for regional or local patients cohorts would be required.

Line 200ff: "Calibration of each scoring system exhibited good performance. The goodness of fit Hosmer-Lemeshow test and p value of each scoring system is shown in Table 4 below." I suspect the listed Chi-Square statistics were calculated according to the 'original' proposed calculation of Ĉ (i.e., with nine degrees of freedom). This should, however, be clearly stated.

Line 215ff: I suspect the calculation of mortality with SAPS 3 was performed using the formulae presented in the original publication. This needs to be emphasised, as the authors themselves point out, that multiple recalibrations exist. More importantly, it is hospital mortality that is calculated - its comparison with ICU mortality is in itself a fallacy.

---

## Round 0.2 · Minor Revisions

1) Please change the wording according to the reviewer's comments.

2) Please elaborate on the model fitting - how and why were the covariables in the multivariable model chosen?

Reviewer 2 ·

Basic reporting

Issues of reporting remain, e.g. "Most of the patients were admitted to the ICU from the emergency department 306(92.5%)" which should probably read "Most of the patients were admitted to the ICU from the emergency department (n=306, 92.5%)" or similar. Changes like these can be performed by the authors at the editors discretion without further need for peer review. Similarly, some final English editing is advised.

Experimental design

The methodology now seems sufficiently described.

Validity of the findings

"Our results highlight the impact of prolonged length of Stay (LoS) in the ICU which is associated with higher adjusted odds of mortality among critically ill patients." I would advise to refrain from such statements without the utilisation of time-to-event utilisation. Moreover, this statement and the analyses leading up to it do not seem to adress the stated research question.

---

## Round 0.3 · accepted · Accept

Dear Dr. Kassam,

Thank you for your submission to PeerJ.
I am writing to inform you that your manuscript - Performance in mortality prediction of SAPS 3 And MPM-III scores among adult patients admitted to the ICU of a private tertiary referral hospital in Tanzania: A retrospective cohort study - has been Accepted for publication.

Congratulations!